# Guinea Pig Sperm Morphology and Fertility under Different Photoperiod

**DOI:** 10.3390/ani13142249

**Published:** 2023-07-09

**Authors:** Hurley Abel Quispe-Ccasa, Yander M. Briceño-Mendoza, Ilse Silvia Cayo-Colca

**Affiliations:** 1Graduate School, Universidad Nacional Toribio Rodríguez de Mendoza of Amazonas, Chachapoyas 01001, Amazonas, Peru; 2Facultad de Ingeniería Zootecnista, Agronegocios y Biotecnología (FIZAB), Universidad Nacional Toribio Rodríguez de Mendoza of Amazonas, Chachapoyas 01001, Amazonas, Peru; yander.briceno@untrm.edu.pe

**Keywords:** *Cavia porcellus* L., illumination, morphometry, acrosome, sperm nucleus, precocity

## Abstract

**Simple Summary:**

Sperm morphology can predict the reproductive potential of males, since the high rate of abnormalities could determine their infertility. In this study, the morphological characteristics of spermatozoa from guinea pigs subjected to different photoperiodic stimuli were determined. The sperm morphology of guinea pigs subjected to a photoperiod by LED light, a photoperiod by sunlight, and a room without a direct light source were compared. In the LED light photoperiod, the spermatozoa had the largest perimeter and nuclear area; in addition, the age of mating and first calving of the females was earlier than in the room without a direct light source. In the photoperiod by sunlight, no pregnancies were achieved. LED light photoperiod can improve the morphological quality of guinea pig sperm and therefore, their reproductive capacity.

**Abstract:**

Sperm morphology can predict the reproductive male fertilizing potential. This study aimed to determine the morphological and morphometric spermatozoa characteristics from guinea pigs subjected to different photoperiodic stimulation. Thirty F1 guinea pigs were randomly assigned to three photoperiodic treatments: FT1 (photoperiod with 10 Light/14 Dark LED light), FT2 (photoperiod with 10L/14D sunlight), and FT0 (room without direct light source). At 107 ± 9.8 days of age, sperm concentration and motility were higher in the FT0 and FT1 groups (*p* < 0.05); furthermore, there were no differences in nucleus length and ellipticity between the FT0 and FT1 groups, but the sperm of the FT1 group was higher in perimeter and nuclear area, while that of the FT0 group was higher in roughness, regularity, midpiece length, and tail (*p* < 0.01). Expanding acrosome (Type 2) was more frequent in the FT2 group, but there was variation in head measurements between all morphological categories. Pregnancy rate, calving age, and mating age were higher in the FT0 group; meanwhile, the FT1 group initiated successful matings earlier (*p* < 0.01). The FT0 group had a higher fertility rate, and the age of mating and first calving were earlier in the FT1 group than the FT0 group, but no pregnancies were reported for the FT2 group. Photoperiodic stimulation can increase the morphometric dimensions of guinea pig spermatozoa, favoring the reproductive characteristics, but sunlight could reduce their size due to heat stress.

## 1. Introduction

Photoperiods regulate key physiological processes in animals, including reproduction, especially in seasonally reproducing species, which is regulated by an endogenous hormonal rhythm [1,2]. In mammals, this regulation may involve the activity of melatonin secreted by the pineal gland only at night [3]. Melatonin receptors are found in many structures and organs, such as the central nervous system, pituitary gland, granulosa cells of pre-ovulatory follicles, spermatozoa, etc. Melatonin concentration is related to the intensity and wavelength of light incident on the retina (high at night and low during the day) [4]. Light incidence on the eye’s retina could trigger a response of the pituitary gland via the pars tuberalis for synchronisation with secretory organs involved in reproductive activity [5,6].

One of the photoperiodic effects is observed in a marked reproductive seasonality of some species, which are found in regions with latitudes further away from the equator. For instance, variations in plasma testosterone concentration, testicular size, seminal production, and sperm concentration have been reported in goats [2] and rams [7]. In higher latitude regions, species limit the birth and rearing to the less rugged season with higher food availability; for this purpose, the photoperiod is a valuable signal, indicating better environmental conditions, initiating ovulation and oestrus. It was reported that in non-seasonal breeding species, e.g., male guinea pigs, a sensitivity to photoperiod is demonstrated, achieving higher body weight and testosterone concentration in a photoperiod of 16 h of light and 8 h of darkness at 25 °C compared to a photoperiod of 8 h of light and 16 h of darkness at 15 °C, showing greater precocity for sexual maturity by early peaks of testosterone in the blood of *Cavia aperea* f. porcellus [8] and *Cavia aperea* [9]. Similarly, in *Cavia aperea* females, increasing photoperiod may stimulate the onset of female reproduction, probably due to its role as a signal of more favorable environmental conditions [10].

Within sperm quality assessment, morphology is a characteristic able to predict the fertilizing capacity of a reproductive male, and a high rate of abnormalities could determine infertility [11,12]. In guinea pigs, the shape of the sperm head has already been classified according to the state of exocytosis of the acrosome [13], and Cabeza et al. [14] determined the dimensions of the head and tail by manual measurements, associating them with reproductive problems. Therefore, this work aims to determine guinea pig spermatozoa’s morphological and morphometric characteristics when subjected to different photoperiodic treatments.

## 2. Materials and Methods

The study was conducted in Chachapoyas province, Amazonas region in the southern hemisphere, during the spring season, from September to December 2020. The area was located at an altitude of 2339 m above sea level, between the coordinates latitude/longitude 6°14′03″ S/77°51′07″ W, in a temperate to cold climate zone, with an annual average temperature of 15 °C and 1578 mm of rainfall (UNTRM-Amazonas Meteorological Station, 2016).

### 2.1. Experimental Design

The experiment was carried out under a split-plot complete randomized design, with three treatments (photoperiod-FT) and five experimental units (EUs) each (EU = 2 animals). The EUs were F1 guinea pigs from a crossbreed of four closely related males with three females, each. The mating and the F1 pups-lactation were carried out under homogeneous lighting (39.3 to 60.8 lux), under photoperiodic conditions of 12 h of light and 12 h of darkness (12L/12D). An initial weighing was performed prior to treatment allocation to verify group homogeneity.

### 2.2. Animal Housing

A 6.0 × 4.5 × 2.5 breeding module made of fiber cement walls, a polypropylene roof, and a curtain-based ventilation system was installed. In four cages of dimensions 1.0 × 1.0 × 0.45 m, four five-month-old male guinea pigs were raised and bred with three females of the same age. All the animals were bought from a nearby commercial farm. Litters of six to eleven pups were born in each cage (mean of 2.9 pups per dam), which were weaned at 21 days. At the age of 26 ± 4 days and an average weight of 262 ± 60 g, thirty guinea pigs (17 males and 13 females) were randomly and paired distributed into fifteen cages of 1.0 × 0.5 × 0.45 m (13 mixed cages with 1 male + 1 female, and 2 with only males), to be housed throughout the experiment. The cages were made of polyethene-coated wire mesh, and the floor of compacted soil. Each cage had a plate feeder and an automatic drinker.

### 2.3. Photoperiods

Three independent rooms of 2.8 × 1.5 × 2.3 m height, each containing five cages, were set up. Each room corresponded to each photoperiod (FT). The first FT had 10 h of LED light (08:00 a.m. to 06:00 p.m.) and 14 h of darkness (10L/14D). The light was generated with a 62-light LED lamp of 3.1 W power and an average light intensity of 175.67 lux at 1.5 m from the source (CN-L862Y, CAFINI, Shenzhen, China) (FT1). The second room had sunlight at an average light intensity of 1254.67 lux at 08:30 a.m. and 476.17 lux at 04:30 p.m. (10L/14D). The entry of the light was managed by curtains opening from 08:00 a.m. to 06:00 p.m. Curtains were located on the eastern and northeastern sides of the shed and 1.2 m above the floor (FT2). The third environment had no direct light source and no photoperiodic schedule (FT0), recording a mean light intensity of 2.36 lux at 08:30 a.m. and 1.17 lux at 04:30 p.m. The study area’s day length during October, November, and December 2020 was 12:10 to 12:30 h of light and 11:50 to 11:30 h of darkness (Meteocast©). In all rooms, ambient temperature and relative humidity were recorded every 10 min with two Thermo hygrometers installed 10 cm above the ground (datalogger; HT71N, PCE Instruments, Meschede, Germany).

Guinea pigs were fed a commercial concentrate based on yellow corn, soybean cake, wheat by-product, molasses, rice flour, rice by-product, micronutrients, vitamins, minerals, and B complex (18% crude protein and 2.8 Mcal/kg); alfalfa forage (*Medicago sativa*); and Guatemala grass (*Tripsacum laxum*). The daily ration of concentrate was 10% of live weight (recalculated weekly) together with green fodder, fed twice daily (08:00 to 09:00 a.m. and 04:00 to 05:00 p.m.). For diarrhea prevention, Daimeton^®^T (Sulfamonomethoxine + Trimethoprim) was added to the morning feed (2 g per 1 kg) and B complex in the drinking water (1 g per liter) for the first three days. Drinking water was provided ad libitum by an automatic waterer.

### 2.4. Sperm Recovery and Analysis

At 107 ± 9.8 days of age, sixteen guinea pigs were slaughtered by transection of the carotid arteries and jugular vein, recording an approximate bleeding time of 30 s. This process was carried out following ARRIVE 2.0 (Animal Research: Reporting of In Vivo Experiments) and the American Veterinary Medical Association (AVMA) euthanasia guidelines [15,16]. Following exsanguination, the testes were removed by a cut on the left side of the scrotal pouch. The organs were placed in polyethylene bags and transported at 37 °C to the Laboratory of the Sperm Collection Centre of the Institute of Livestock and Biotechnology (IGBI—UNTRM). The caudal epididymis side was sectioned into a Petri dish on a 37 °C heat plate. The Petri dish contained 0.4 mL of tempered Tris medium (3.028 g molecular grade Tris base, 1.7 g citric acid, and 1.25 g D-fructose in 100 mL of distilled water) [17], pH 7.0 and osmolarity 300 mOsm/L. Slight cuts were made to promote the outflow of spermatozoa into the medium [18]. Then, 0.6 mL of the same Tris medium was added, recovered in 1.5 mL microtubes, and kept at 37 °C. The concentration and sperm motility was analyzed with a Makler chamber under a phase contrast microscope (CX31, OLYMPUS, Tokyo, Japan), in a Sperm Class Analyzer (SCA^®^ 5.3.0.0. VET edition, Microptic, Barcelona, Spain) with camera (acA780-75gc, BASLER, Ahrensburg, Germany). The percentages of progressive, non-progressive and total motility were determined, based on a mean of approximately 990 cells in 5 fields captured per sample.

### 2.5. Sperm Morphology

For morphology analysis, a smear was made with a Hemacolor^®^ blood cell differentiation stain (Merck, Darmstadt, Germany) according to the manufacturer’s protocol. Briefly, the smear was immersed in Fixation Solution I (methanol) five times for one second each, then in Solution II (eosin) three times for one second each, then in Solution III (Azure) six times for one second each, and finally, in Wash Solution (pH 7.2) with two immersions of 10 s each, and then dried vertically at room temperature. For analysis, 50 spermatozoa per sample were captured at 100× with immersion oil, with a microscope (BX53, OLYMPUS, Tokyo, Japan) equipped with a camera (U-TV0.63XC, OLYMPUS, Tokyo, Japan). Cells were morphologically classified into four categories according to the state of acrosome exocytosis [13]: intact acrosome (1), expansion by acrosomal reaction (2), the onset of decomposition (3), and loss of acrosomal contents and matrix (4) (Figure 1). In addition, atypical shapes such as micronucleus, macronucleus, pyriform, elongated, and lanceolate were counted (Figure 2). Subsequently, the nucleus (length L, width W, area A and perimeter P), acrosome area and perimeter, head length, midpiece length and width, and tail length were measured (CellSens, OLYMPUS, Tokyo, Japan). In addition, the dimensionless parameters of ellipticity (L/W), roughness (4πA/P^2^), elongation ([L − W]/[L + W]), and regularity (πLW/4A) of the sperm nucleus were calculated.

### 2.6. Fertility

The twelve EUs consisted of one male and one female each. At the time of first calving, age at calving, probable age at mating (considering a mean of 68 days of gestation), pregnancy percentage, and number of offspring per calving in each FT room were determined.

### 2.7. Statistical Analysis

The normal distribution and homogeneity of variances were verified for live weight, testicular weight, sperm concentration, progressive motility, non-progressive motility, and total motility. The morphometric parameters did not meet the assumptions of normality (Kolmogorov–Smirnov test) and homogeneity of variances (Levene’s test), which is why they were analyzed with a non-parametric Kruskal–Wallis test (*p* < 0.05) for more than two groups and a Mann–Whitney U test (*p* < 0.05) for two groups. The age at mating, age at calving, and the number of offspring were evaluated with ANOVA and pregnancy rate with Chi-square test. Correlations were analyzed with Pearson’s coefficients in SPSS v.15.0 software (SPSS Inc., Chicago, IL, USA).

## 3. Results

The analysis of sixteen adult male guinea pigs of 107 ± 9.8 days of age showed non-significant differences in live and testicular weight between FT treatments (*p* > 0.05), but the sperm concentrations of the FT0 and FT1 groups were significantly higher than that of the FT2 group (*p* < 0.05). Similarly, the sperm of the FT0 and FT1 groups showed higher non-progressive motility and total motility compared to the FT2 group (*p* < 0.05) (Table 1).

The morphometric variables of the nucleus, acrosome, midpiece, and tail were measured in 938 spermatozoa from 16 guinea pigs. Differences (*p* < 0.01) were found in most parameters between FTs, except nucleus width (*p* > 0.05) (Table 2, Appendix A). The length, ellipticity, and elongation of the nucleus of the FT0 and FT1 groups’ spermatozoa did not differ from each other but were greater than those of the FT2 group. The FT1 group’s spermatozoa had a larger nuclear perimeter and area than the other groups, but the midpiece length, and tail length of the FT0 group’s spermatozoa were significantly larger than those of the other groups. The roughness and regularity of the nucleus differed between groups, being greater in the FT0 group, and the midpiece width, which was significantly larger in the FT2 group.

In addition, 6950 spermatozoa were morphologically classified according to the acrosome status and the nucleus abnormalities (micronucleus, macronucleus, pyriform, elongated, and lanceolate) (Table 3 and Appendix A). The rate of nucleus abnormalities ranged from 0 to 8.95%, giving a high coefficient of variation; however, we found no significant differences between groups (*p* > 0.05), and means did not reach 2% of abnormalities. We did not find significant differences (*p* > 0.05) between Type 1, Type 3, and Type 4 spermatozoa among the FTs, but we did find a higher frequency of Type 2 morphology in FT2 guinea pigs (*p* < 0.05).

Significant differences were found in most morphometric parameters, except nucleus roughness, midpiece width, and tail length (*p* > 0.05). Larger nucleus dimensions were observed in Type 4 spermatozoa and smaller in Type 1 spermatozoa. The arrangement of the acrosome is the basis of the morphological classification; therefore, larger perimeters, acrosomal areas, and head lengths were observed in Type 2 spermatozoa (expanded acrosome), followed by Type 1, Type 3, and finally, Type 4 (no acrosomal structure and therefore, shorter head length) (*p* < 0.01) (Table 4).

According to Pearson’s coefficient, sperm classification did not correlate with live weight, testicular weight, concentration, and motility. Type 1 correlated negatively with Type 3 and Type 4, and Type 3 correlated positively with the highest frequency of sperm nucleus abnormalities (Table 5).

During the FT treatment period, each male guinea pig was housed with a female of the same age to homogenize sexual stimulation. At the end of the study, no FT2 females (0/5), all FT0 females (4/4), and half of the FT1 females (2/4) were pregnant. The Chi-square test showed a significant association of pregnancy rate with FT (*p* < 0.05), being higher in the FT0 group, followed by the FT1 group. The average number of offspring of pregnant females was higher in the FT1 group, but not significantly (*p* > 0.05). However, age at parturition and probable age at mating in the FT0 group was significantly later than in the FT1 group (*p* < 0.01), showing a higher reproductive earliness in FT1 guinea pigs, with a difference of approximately 23 days (Table 6).

Fertility variables were correlated with sperm characteristics, reported in Table 7. The pregnancy rate was correlated with progressive, non-progressive, and total motility. On the other hand, the number of offspring was positively correlated with live weight and non-progressive motility; probable age at mating was negatively correlated with testicular weight, non-progressive motility, and type 4 spermatozoa (without acrosome).

## 4. Discussion

This study evaluated the effect of three different light exposures (photoperiod with LED light (FT1), photoperiod with sunlight (FT2), and no direct light exposure (FT0)) on guinea pigs’ sperm morphology. Reports showed that although there was no effect of FTs on live and testicular weights, male guinea pigs subjected to an 8L/16D photoperiod versus a 16L/8D photoperiod achieved lower weights, as well as inhibition of somatic and testicular growth in rats under the same photoperiod conditions [8,19]. The increased testicular growth could be related to the higher blood testosterone concentrations in seasonally and non-seasonally reproducing species influenced by photoperiod [2,7,8,9,20].

Guinea pig spermiogram studies show variable concentrations and motility rates depending on the method of extraction and analysis. In spermatozoa from the epididymis, as in this study, Ayala Guanga et al. [21] reported 418.0 × 10^6^/mL and 58% motility, Rodríguez et al. [22] reported 95% motility, and Ferdinand et al. [23] reported 149.85 × 10^6^/mL sperm concentration. Using electroejaculation, the values are generally low, ranging from 36.70 to 47.33 × 10^6^/mL for concentration and 49.49 to 90.86% for individual motility [14,24,25].

Live weight, testicular weight, and sperm concentration in a photoperiod under LED light were higher than in a room without direct light. The positive effect of photoperiod on testosterone concentration, testicular volume, and sperm concentration in seasonal species is known [2,7], due to the greater secretion of melatonin [4,26]. The presence of melatonin receptors in the reproductive organs and spermatozoa [1] has been reported, and this hormone could signal the appropriate environmental conditions for reproduction in mammals. In mammals, the eye’s retina is the photoreceptor that receives information about the presence of light and darkness. This activates the circadian rhythm in the pineal gland for melatonin secretion during the absence of light. Melatonin activity controls the secretion of TSH in the pars tuberalis of the pituitary gland for the release of gonadotropins. In males, LH is active in Leyding cells for testosterone production, and FSH is active in Sertoli cells for inhibin production, which regulates spermatogenesis [6]. Guinea pigs may be sensitive to daylength, as light stimulation with prolonged photoperiods of 14–16 h of light were able to anticipate puberty in males, evidenced by earlier serum testosterone peaks than in an 8-h light regimen [8,10]. In this study, successful matings and pregnancies occurred earlier in the FT1 group (10L/14D per LED light) than in the FT0 group. In the FT2 group (10L/14D by sunlight), sperm concentration and sperm motility were lower. It is unknown if a photoperiod with LED light in FT1 stimulates greater melatonin production than in FT2, but in the FT2 group, the incidence of sunlight generated a greater increase in environmental temperature, which could negatively impact spermatogenesis.

In the sunlight photoperiod (FT2), the curtains remained open from 08:00 to 18:00 h, to allow light to enter. However, the ambient temperature reached higher peaks (33.8 °C) than in FT1 and FT0 (32.75 °C and 32.65 °C, respectively), due to the fact that the incidence of sunlight in FT2 was from 08:00 to 10:00 a.m. in 20% of the surface of each cage. The guinea pigs in the FT2 group could have experienced frequent episodes of heat stress caused by the temperature generated by solar radiation, which would explain the reduction in spermiogram quality. High housing temperatures could generate heat stress and negative impacts on spermatogenesis, sperm concentration, motility, rate of morphological abnormalities, and sperm DNA damage in guinea pigs [27,28] and rabbit sperm in vitro [29]. The midday temperature peaks in FT2 exceeded the optimal scrotal temperature level of 32.5 °C; therefore, they were probably capable of compromising sperm functionality.

The average values of the morphometric traits length, width, perimeter and nucleus area, roughness, and regularity of guinea pig spermatozoa in this study are higher than those reported by Cabeza et al. [14] by electroejaculation, using Spermac^®^ staining (Minitüb GmbH, Tiefenbach, Germany) and Motic Image Plus v.2.0 software (Motic, Vacouver, Canada); and Yucra [30] in spermatozoa from epididymis, stained with Diff-Quick and analyzed in an ISAS^®^ CASA equipment (Zoitech, Madrid, Spain). However, lower values were obtained for ellipticity, elongation, midpiece length, and tail length than in the literature. The differences may be due to the staining method and measurement determination, since staining can influence actual sperm morphometry, due to dehydration or osmosis of the cells [31,32], but this procedure is essential to improve contrast and make visible the shape of the fixed spermatozoon, for morphometric analysis [33,34].

In the FT2 group, the smallest spermatozoa were found, except for the width of the intermediate piece. Heat stress could have altered spermatogenesis in FT2 guinea pigs, increasing the frequency of microcephalics. While, Malo et al. [35] argue that a bulky midpiece and less head elongation in the spermatozoon could hinder motility, making it slower and less progressive, especially considering that the nucleus and head of guinea pig spermatozoa are less elongated than in other species. The photoperiodic stimulation in FT1 could have influenced spermatogenesis due to the greater melatonin and testosterone activity, obtaining sperm with a greater perimeter and nuclear area and other sperm variables [4]. In the spermatozoa of the FT0 group, with greater midpiece and tail length, more energy could be generated from mitochondria in these structures, and consequently, be positively associated with adequate swimming speed and motility [34,35].

Regarding the morphological classification of guinea pig sperm based on the exocytosis state of the acrosome [13] (Type 1: intact acrosome, Type 2: expansion due to an acrosome reaction, Type 3: initiation of decomposition, and Type 4: loss of acrosomal content), a higher frequency of Type 1 spermatozoa was found in the FT1 and FT0 groups, and Type 2 in the FT2 group. Guinea pig sperm are morphologically homomorphic due to the homogeneity of the sperm, especially at the level of the nucleus [14,36]. The morphometric traits varied significantly between morphological groups, with the exception of core roughness, midpiece width, and tail length. The largest area and acrosomal perimeter correspond to Type 2, due to the expansion of the acrosome. Type 4 showed a larger nucleus size than Types 1, 2, and 3, in which the presence of the acrosome could have influenced staining and manual measurement of the nucleus. Hemacolor^®^ staining could co-stain the overlapping nucleus and acrosome, underestimating nucleus measurements. Differential staining techniques may be useful in future morphological studies.

Age at mating and age at calving were lower in the FT1 group, and pregnancy rate was higher in the FT0 group. This 23-day difference in mating age between the two groups is a finding consistent with Bauer et al. [8] and Guenther et al. [9], who found greater sexual precocity (precocious puberty and early blood testosterone peaks) in guinea pigs subjected to a photoperiod of more daylight hours. The period of lower incidence of light stimulates the secretion of melatonin, and it is known that the greater activity of the pineal gland could delay puberty and sexual maturity in males [37]. This advance in the precocity of males and females by photoperiodic stimulation could constitute an improvement in the productivity of the farms since, generally, three-month-old females and four-month-old males are used for reproductive purposes. According to literature reports, the average number of guinea pig pups per birth is three. In this experimental study, successful early matings with two to four pups were achieved at approximately two months of age using a 10L/14D LED photoperiod. In the FT0 group, births with one to two pups were observed, without significant differences between groups. A greater number of matings, in future studies, could clarify the effect of photoperiod on guinea pig litter size. However, this photoperiodic stimulation could be included in small farms to reduce the intergeneration time and increase the number of births per year. The absence of pregnancies in the FT2 group may be associated with low sperm motility and the effects generated by heat stress.

## 5. Conclusions

Stimulation with a photoperiod by 10L/14D LED light can improve some quality characteristics and morphometric traits of guinea pig spermatozoa. Whereas, a 10L/14D photoperiod of sunlight could generate higher peaks in environmental temperature, heat stress, and consequently, low spermiogram values and reduced spermatozoa. Pregnancy rate was not improved in a daylight period with LED light compared to a room without direct light, but mating and calving were earlier, up to 23 days. In a photoperiod of 10L/14D sunlight, no pregnancies were obtained.

## Figures and Tables

**Figure 1 animals-13-02249-f001:**
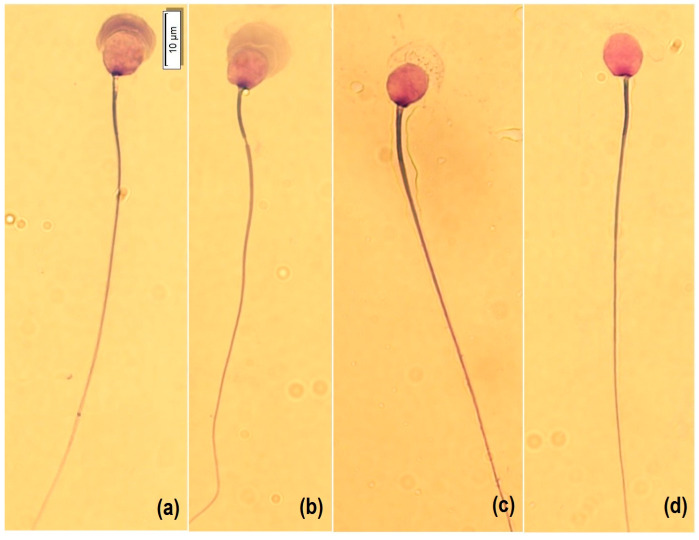
Morphology of guinea pig spermatozoa with Hemacolor^®^ staining (Merck, Darmstadt, Germany). Type 1: intact acrosome (**a**), Type 2: expansion by acrosomal reaction (**b**), Type 3: the onset of decomposition (**c**), and Type 4: loss of acrosomal contents and matrix (**d**). Bar represents 10 μm.

**Figure 2 animals-13-02249-f002:**
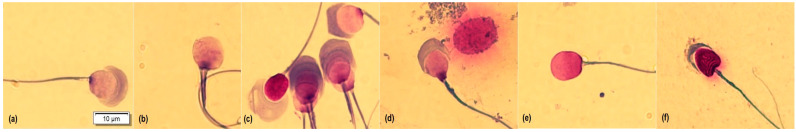
Atypical forms in guinea pig spermatozoa with Hemacolor^®^ staining (Merck, Darmstadt, Germany). Normal (**a**), spermatozoa with macronucleus (**b**), micronucleus (**c**), pyriform (**d**), elongated (**e**), and lanceolate (**f**). Bar represents 10 μm.

**Table 1 animals-13-02249-t001:** Sperm analysis of guinea pigs (107 ± 9.8 days of age) subjected to photoperiodic treatment.

Variable	FT0	FT1	FT2	*p* Value
N	6	5	5	
Live weight (g)	1163.18 ± 66.17	1264.38 ± 48.73	1036.82 ± 83.05	0.14
Testicle weight (g)	12.24 ± 1.16	14.72 ± 1.29	13.70 ± 2.14	0.60
Concentration (M/mL)	915.23 ± 154.97 ^ab^	1151.10 ± 144.26 ^a^	456.19 ± 146.57 ^b^	0.04 *
Progressive motility (%)	29.44 ± 5.59	22.88 ± 5.86	15.61 ± 5.03	0.27
Non progressive motility (%)	50.19 ± 2.67 ^a^	48.49 ± 10.63 ^a^	13.89 ± 10.25 ^b^	0.02 *
Total motility (%)	79.63 ± 3.12 ^a^	71.37 ± 15.03 ^a^	29.50 ± 12.70 ^b^	0.01 *

Mean ± Standard Error. FT0: No direct light stimulation, FT1: Photoperiod with LED light 10 h Ligth/14 h Darkness (10L/14D), FT2: Sunlight Photoperiod 10L/14D. M/mL: Millions of sperm per milliliter. Different superscript letters in rows (^a,b^) indicate significant differences. *: Significant differences at level *p* < 0.05. There are not enough arguments to establish differences in the sperm concentration of FT0 with respect to FT1 and FT2 (^ab^).

**Table 2 animals-13-02249-t002:** Morphometric analysis of guinea pig spermatozoa subjected to photoperiodic treatment.

Traits	FT0	FT1	FT2	K-W Test
N	314	280	344	
Core length (µm)	8.25 ± 0.02 ^a^	8.25 ± 0.02 ^a^	8.16 ± 0.02 ^b^	<0.01 **
Core width (µm)	7.34 ± 0.02	7.37 ± 0.02	7.36 ± 0.02	0.83
Core perimeter (µm)	26.10 ± 0.05 ^b^	26.40 ± 0.05 ^a^	26.21 ± 0.06 ^b^	<0.01 **
Core area (µm^2^)	48.93 ± 0.20 ^b^	49.94 ± 0.19 ^a^	49.16 ± 0.23 ^b^	<0.01 **
Ellipticity	1.125 ± 0.004 ^a^	1.121 ± 0.003 ^a^	1.110 ± 0.003 ^b^	<0.01 **
Rugosity	0.901 ± 0.001 ^a^	0.899 ± 0.001 ^b^	0.898 ± 0.001 ^c^	<0.01 **
Elongation	0.058 ± 0.002 ^a^	0.056 ± 0.001 ^a^	0.051 ± 0.001 ^b^	<0.01 **
Regularity	0.972 ± 0.001 ^a^	0.957 ± 0.001 ^c^	0.961 ± 0.001 ^b^	<0.01 **
Acrosome perimeter (µm)	33.83 ± 0.16	33.98 ± 0.19	33.56 ± 0.23	0.05
Acrosome area (µm^2^)	79.11 ± 0.79	79.92 ± 0.95	77.98 ± 1.18	0.05
Head length (µm)	12.25 ± 0.04	12.31 ± 0.06	12.31 ± 0.09	0.68
Midpiece length (µm)	11.45 ± 0.03 ^a^	11.17 ± 0.04 ^b^	11.14 ± 0.05 ^b^	<0.01 **
Midpiece width (µm)	0.634 ± 0.005 ^c^	0.653 ± 0.006 ^b^	0.715 ± 0.008 ^a^	<0.01 **
Tail length (µm)	89.66 ± 0.64 ^a^	88.72 ± 0.47 ^b^	89.43 ± 0.26 ^b^	<0.01 **

Mean ± Standard Error. FT0: No direct light stimulation, FT1: Photoperiod with LED light 10L/14D, FT2: Sunlight Photoperiod 10L/14D. Different superscript letters in rows (^a,b,c^) indicate significant differences. **: Significant difference at level *p* < 0.01 by Kruskal-Wallis (K-W) test.

**Table 3 animals-13-02249-t003:** Morphological classification of guinea pig spermatozoa according to acrosomal status, subjected to photoperiodic treatment.

Category	FT0	FT1	FT2	*p* Value
N	2903	2043	2004	-
Type 1 (%)	62.22 ± 8.63	47.67 ± 5.31	38.87 ± 9.88	0.06
Type 2 (%)	7.27 ± 2.92 ^a^	11.53 ± 3.09 ^ab^	26.16 ± 10.56 ^b^	0.04 *
Type 3 (%)	11.81 ± 3.78	11.97 ± 1.89	9.67 ± 1.58	0.62
Type 4 (%)	18.70 ± 4.26	28.83 ± 5.73	25.30 ± 10.27	0.50
Abnormal (%)	1.84 ± 1.69	1.64 ± 0.55	1.50 ± 0.50	0.81

Mean ± Standard Error. FT0: No direct light stimulation, FT1: Photoperiod with LED light 10L/14D, FT2: Sunlight Photoperiod 10L/14D. Different superscript letters in rows (^a,b^) indicate significant differences at level *p* < 0.05 (*). There are not enough arguments to establish differences in the frequency of Type 2 sperm morphology of FT1 with respect to FT0 and FT2 (^ab^).

**Table 4 animals-13-02249-t004:** Morphometric traits of guinea pig spermatozoa according to morphological classification.

Traits	Type 1	Type 2	Type 3	Type 4	K-W Test
N	447	194	101	196	
Core length (µm)	8.211 ± 0.019 ^ab^	8.170 ± 0.025 ^b^	8.199 ± 0.036 ^ab^	8.279 ± 0.029 ^a^	0.03 *
Core width (µm)	7.301 ± 0.016 ^b^	7.424 ± 0.020 ^a^	7.375 ± 0.032 ^a^	7.406 ± 0.025 ^a^	<0.01 **
Core perimeter (µm)	26.081 ± 0.048 ^b^	26.342 ± 0.061 ^a^	26.313 ± 0.090 ^ab^	26.426 ± 0.081 ^a^	<0.01 **
Core area (µm^2^)	48.730 ± 0.178 ^b^	49.709 ± 0.227 ^a^	49.578 ± 0.341 ^ab^	50.131 ± 0.307 ^a^	<0.01 **
Ellipticity	1.126 ± 0.003 ^a^	1.102 ± 0.004 ^b^	1.113 ± 0.006 ^ab^	1.119 ± 0.004 ^a^	<0.01 **
Rugosity	0.899 ± 0.0004	0.899 ± 0.001	0.899 ± 0.001	0.901 ± 0.001	0.15
Elongation	0.059 ± 0.001 ^a^	0.048 ± 0.002 ^b^	0.053 ± 0.003 ^ab^	0.056 ± 0.002 ^a^	<0.01 **
Regularity	0.967 ± 0.001 ^a^	0.959 ± 0.002 ^b^	0.959 ± 0.002 ^b^	0.962 ± 0.002 ^b^	<0.01 **
Acrosome perimeter (µm)	33.283 ± 0.100 ^b^	35.209 ± 0.278 ^a^	29.629 ± 0.617 ^c^	-	<0.01 **
Acrosome area (µm^2^)	76.446 ± 0.480 ^b^	86.124 ± 1.460 ^a^	58.584 ± 2.651 ^c^	-	<0.01 **
Head length (µm)	12.105 ± 0.032 ^b^	12.766 ± 0.080 ^a^	11.212 ± 0.294 ^c^	8.279 ± 0.029 ^d^	<0.01 **
Midpiece length (µm)	11.297 ± 0.036 ^a^	11.318 ± 0.043 ^a^	11.254 ± 0.065 ^ab^	11.086 ± 0.062 ^b^	<0.01 **
Midpiece width (µm)	0.665 ± 0.006	0.673 ± 0.009	0.685 ± 0.014	0.666 ± 0.008	0.41
Tail length (µm)	89.201 ± 0.475	88.715 ± 0.713	89.342 ± 0.465	90.011 ± 0.288	0.06

Mean ± Standard Error. Different superscript letters in rows (^a,b,c,d^) indicate significant differences. *: Significant difference at level *p* < 0.05. **: Significant difference at level *p* < 0.01 by Kruskal–Wallis (K-W) test. There are not enough arguments to establish differences of traits between groups (^ab^).

**Table 5 animals-13-02249-t005:** Pearson’s correlation for morphological classification and sperm variables of guinea pigs.

	Type 1	Type 2	Type 3	Type 4	Abnormalities
Live weight	0.00	0.00	0.08	−0.03	−0.07
Testicular weight	−0.24	0.46	0.10	−0.11	−0.19
Concentration	0.22	−0.01	0.00	−0.28	0.14
Progressive motility	0.29	−0.07	−0.16	−0.26	0.06
Non Progressive motility	−0.07	0.08	0.23	−0.08	0.05
Total motility	0.07	0.03	0.11	−0.17	0.06
Type 1		−0.61 *	−0.38	−0.56 *	−0.21
Type 2			0.11	−0.26	0.25
Type 3				−0.01	0.63 *
Type 4					−0.24

*: Significant correlation at level *p* < 0.05 (two-sided).

**Table 6 animals-13-02249-t006:** Fertility of guinea pigs under different light exposure, subjected to photoperiodic treatment.

Variable	FT0	FT1	FT2	*p* Value
N	4	4	5	
Pregnancy rate (%)	100.0 ^a^	50.0 ^b^	0.0 ^c^	0.01 *
NO. average offspring	1.5 ± 0.3	3.0 ± 1.0	-	0.13
Calving age (days)	152.5 ± 0.6 ^b^	129.5 ± 0.5 ^a^	-	<0.01 **
Probable mating age (days)	84.5 ± 0.6 ^b^	61.5 ± 0.5 ^a^	-	<0.01 **

Mean ± Standard Error. FT0: No direct light stimulation, FT1: Photoperiod with LED light 10L/14D, FT2: Sunlight Photoperiod 10L/14D. Different superscript letters in rows (^a,b,c^) indicate significant differences. *: Significant difference at level *p* < 0.05; **: Significant difference at level *p* < 0.01.

**Table 7 animals-13-02249-t007:** Pearson’s correlation for fertility and sperm variables of guinea pigs.

	Pregnancy Rate	No. Average Offspring	Mating Age
Live weight	0.38	0.83 *	−0.71
Testicular weight	−0.12	0.65	−0.99 **
Concentration	0.33	0.70	−0.19
Progressive motility	0.65 *	−0.54	−0.01
Non Progressive motility	0.67 *	0.88 *	−0.82 *
Total motility	0.73 **	−0.10	−0.51
Type 1	0.25	−0.33	0.63
Type 2	−0.43	−0.21	−0.15
Type 3	0.06	0.08	−0.12
Type 4	0.08	0.57	−0.83 *
Abnormalities	0.10	−0.55	0.37

**: Significant correlation at level *p* < 0.01 (two-sided); *: Significant correlation at level *p* < 0.05 (two-sided).

## Data Availability

The data presented in this study are available on request from the corresponding author. The data are not publicly available due to privacy restrictions.

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
