# Peer review of "Guinea Pig Sperm Morphology and Fertility under Different Photoperiod"

_animals, 2023, doi:10.3390/ani13142249_

Round 1
Reviewer 1 Report
Photoperiod plays important role in animal reproduction, and relates with seasonal estrous of female and spermatogenesis in male. This manuscript was interested and determined the effects of different photoperiodic stimuli on the morphological characteristics of sperm from guinea pigs. Some issue should be addressed before accepted for publication.
Major issues:
1. Photoperiod and temperature are two key regulators of animal reproduction, especially on estrous. In this study, the ambient temperature reached over 33.8 °C which may cause heat stress and affect spermatogenesis, especially in FT2. This can be explained why lower pregnancy rate, number of offspring were observed in FT2 treatments. So, ambient temperature is a key factor and should be included. Authors need to confirm that sperm abnormal morphological characteristics was induced by photoperiod or temperature.
2. The sample number of this study was a relatively smaller.
Minor issues:
1. Line 119: “12:10 to 12:30 hours of light and 11:50 to 11:30 hours of darkness”.
2. In statistical analysis, all P value should be italic format.
3. Table 1-4: the data can be presented by Mean ± SEM. “p-valor” in table 1 should be “p value”.
4. “N° offspring” in table 6 should be “NO. offspring”.
4. Line 289-292 can be deleted.
minor spell issue can be corrected.
Reviewer 2 Report
The article “Guinea pig sperm morphology and fertility under different 2 photoperiod” analyzes the prediction of reproductive potential by observing the morphology of guinea pig sperm exposed to different photoperiods with sunlight, LED-emitted light or no photoperiod, leaving the group in the dark. The photoperiods used were 10/14 hours light-dark; Aspects of feeding, temperature and humidity were controlled throughout the experiment. Sperm obtained from the caudal epididymis were analyzed for their morphology and motility. The morphological descriptions are very detailed and many significant differences between the different photoperiods have been identified. Interesting are the fertility results in the photoperiods without light or artificial LED light increasing the number of pregnant females, early calving and high average offspring with LED light.
Minor concerns:
Title: “Guinea pig sperm morphology and fertility under different 2 photoperiod”
“Guinea pig sperm morphology and fertility under different 2 photoperiods”
2. It is mentioned that the sperm were collected in a Tris buffer containing citric acid and D-fructose, the pH of this buffer is not mentioned, then it is also mentioned that Tris medium was added. Was culture medium used? Or refers to the same buffer?
3. Regarding motility, it does not describe how to methodically analyze motility when including progressive, non-progressive and immobile motility. This must be clearly described in the methodology, stating how many sperm per field were counted in total and from how many animals.
4. Regarding fertility, clarify in the methodology whether the females caged with the males were exposed to the same photoperiod conditions or whether the males were removed from that photoperiod for their reproduction, since in females their fertility is also affected by the photoperiod conditions.
5. In Table 6, the number of offspring is the result of the average number of young out of the total number of pregnant females? Clear it on the table.
6. In terms of fertility, it would be interesting to know the average number of pubs per female in this specie, as the number of mating experiments is limited to compare how photoperiod affects the offspring. Therefore, the difference in the number of pups in FT1 compared to FT0 is not mentioned as the mean is lower in the latter, although the caged females are 100% pregnant.
7. The results are very interesting in terms of how photoperiod affects the morphology, motility and fertility of guinea pigs. It would be interesting to know how this period affects the pineal gland and how it relates to the hormonal signals that favor spermatogenesis and fertility. The authors mention that melatonin may be elevated, which could be a good predictor of photoperiod exposure. However, they do not mention whether LED light stimulates melatonin production more than sunlight.
Round 2
Reviewer 1 Report
The authors respond positively to my comments. I have no any suggestions and comments.